# Utilizing Headspace–Gas Chromatography–Ion Mobility Spectroscopy Technology to Establish the Volatile Chemical Component Fingerprint Profiles of *Schisandra chinensis* Processed by Different Preparation Methods and to Perform Differential Analysis of Their Components

**DOI:** 10.3390/molecules29245883

**Published:** 2024-12-13

**Authors:** Yiping Yan, Bowei Sun, Mengqi Wang, Yanli Wang, Yiming Yang, Baoxiang Zhang, Yining Sun, Pengqiang Yuan, Jinli Wen, Yanli He, Weiyu Cao, Wenpeng Lu, Peilei Xu

**Affiliations:** 1Institute of Special Animal and Plant Sciences, Chinese Academy of Agricultural Sciences, Changchun 130112, China; 82101225211@caas.cn (Y.Y.); 2022050841@ybu.edu.cn (B.S.); 82101232246@caas.cn (M.W.); wangyanli@caas.cn (Y.W.); yangyiming@caas.cn (Y.Y.); zhangbaoxiang@caas.cn (B.Z.); 82101225210@caas.cn (Y.S.); 82101222242@caas.cn (P.Y.); 82101215188@caas.cn (J.W.); 82101215184@caas.cn (Y.H.); 82101202231@caas.cn (W.C.); 2College of Agriculture, Yanbian University, Yanji 133002, China; 3Jilin Provincial Key Laboratory of Traditional Chinese Medicinal Materials Cultivation and Propagation, Changchun 130112, China

**Keywords:** *Schisandra chinensis*, HS-GC-IMS, traditional Chinese medicine processing, volatile components, OPLS-DA

## Abstract

In order to characterize the volatile chemical components of *Schisandra chinensis* processed by different Traditional Chinese Medicine Processing methods and establish fingerprint profiles, headspace–gas chromatography–ion mobility spectrometry (HS-GC-IMS) technology was employed to detect, identify, and analyze *Schisandra chinensis* processed by five different methods. Fingerprint profiles of volatile chemical components of *Schisandra chinensis* processed by different methods were established; a total of 85 different volatile organic compounds (VOCs) were detected in the experiment, including esters, alcohols, ketones, aldehydes, terpenes, olefinic compounds, nitrogen compounds, lactones, pyrazines, sulfur compounds, thiophenes, acid, and thiazoles. Principal component analysis (PCA), Orthogonal Partial Least Squares Discriminant Analysis (OPLS-DA), and Pearson correlation analysis methods were used to cluster and analyze the detected chemical substances and their contents. The analysis results showed significant differences in the volatile chemical components of *Schisandra chinensis* processed by different methods; the Variable Importance in Projection (*VIP*) values of the OPLS-DA model and the *P* values obtained from one-way ANOVA were used to score and screen the detected volatile chemical substances, resulting in the identification of five significant chemical substances with the highest *VIP* values: Alpha-Farnesene, Methyl acetate,1-octene, Ethyl butanoate, and citral. These substances will serve as marker compounds for the identification of *Schisandra chinensis* processed by different methods in the future.

## 1. Introduction

Traditional Chinese Medicine Processing is a traditional method of processing most herbs and medicinal materials in China, widely used in traditional Chinese medicine. The main purpose of processing is to reduce toxic substances in certain medicinal materials or increase pharmacologically active substances. Processing can also alter the properties of medicinal materials to make them easier to digest or swallow [1,2].

*Schisandra chinensis* is a climbing plant of the *Magnoliaceae* family, and its fruit, “Northern Schisandra”, is a traditional herbal medicine in Asia, also considered a palatable fruit and condiment [3]. *Schisandra chinensis* contains natural chemical constituents with medicinal activity such as lignans [4]. Lignans have been proven to possess hepatoprotective effects [5]. Qu Zhongyuan compared steamed and raw Schisandra chinensis in the treatment of allergic asthma symptoms and, using a rat model, validated that wine-steamed *Schisandra chinensis* processed by Traditional Chinese Medicine Processing has better pharmacological effects than raw Schisandra chinensis [6]. *Schisandra chinensis* has been proven to effectively prevent cell damage caused by UV radiation and effectively inhibit the spread of skin cancer [7]; its volatile oil components have antidepressant effects [8]. *Schisandra chinensis* polysaccharides have been shown to alleviate Parkinson’s disease by activating the autophagy signal MCL-1 [9]; the main constituent of Schisandra chinensis, schisandrin A, can alleviate non-alcoholic fatty liver disease [10]. Processing can increase the medicinally active substances in *Schisandra chinensis* or promote the absorption of active substances by the human body. Several Chinese scientists have conducted experiments on the widely recognized lignan medicinal components of *Schisandra chinensis*, proving that different processing methods significantly affect the content of various lignans in *Schisandra chinensis*. Ge conducted a systematic study evaluating the effects of vinegar-processed *Schisandra chinensis* on acute liver injury. The results showed that vinegar processing not only identified the main active components for treating acute liver injury but also isolated the by-product 5-Hydroxymethylfurfural (5-HMF), produced during processing due to the Maillard reaction, which has a therapeutic effect on liver injury, indicating that processing can increase the active substances in *Schisandra chinensis* and enhance its efficacy [11]. Dong studied the addition of salt to the processing of compound medicines with *Schisandra chinensis* as the main medicinal material. By examining the changes in chemical components before and after processing, it was verified that adding salt during processing can improve the total apoptosis rate of spermatogenic cells and increase the activity of compound medicines [12].

The characterization of volatile chemical substances in herbal medicines has received increasing attention in recent years. Yan identified the gas-phase components of *Schisandra chinensis* fruits and branch sap of different colors and established fingerprint profiles for *Schisandra chinensis* of different colors [13]. Although significant differences exist in the compositional profiles resulting from various preparation methods for the same medicinal plant, these differences are often not discernible through visual or olfactory cues after the herbs have been processed and dried. Traditional methods of distinguishing *Schisandra chinensis* preparations involve assessing color and odor. However, it has been observed that the color of *Schisandra chinensis* fruit can vary considerably, not only due to genetic diversity but also in response to harvest timing. In practical production applications, relying solely on color is insufficient for distinguishing *Schisandra chinensis* preparations. Furthermore, the aroma of processed *Schisandra chinensis* tends to diminish during the drying process, making accurate differentiation based on smell challenging. Therefore, it is essential to develop a non-destructive and rapid method for assessing volatile component differences and establish volatile compound fingerprint profiles for different *Schisandra chinensis* preparations. Several research teams have used gas chromatography–mass spectrometry (GC-MS), gas chromatography–ion mobility spectrometry (GC-IMS), and other volatile chemical substance detection methods to identify and establish fingerprint profiles for various Traditional Chinese Medicine Processing products: Xing characterized the volatile chemical components of Polygonum multiflorum processed by different methods using GC-MS and GC-IMS technologies, identifying characteristic substances of the processed products [14]; Fu used GC-IMS technology to identify the chemical components of nine processed tangerine peels, analyzing the volatile chemical components at different processing stages in detail [15]; Gao used GC-IMS technology to study different processing methods of pomegranate seeds, and the results showed the fingerprint profiles and key identification substances of processed pomegranate seeds [16].

Headspace–gas chromatography–ion mobility spectrometry (HS-GC-IMS) technology is a newly emerging sensitive and rapid gas-phase separation detection technology of recent years. This technology can quickly and accurately distinguish and identify volatile substances [17,18]. It combines ion mobility spectrometry and gas chromatography analysis methods, producing a two-dimensional spectrum of volatile compounds. This is based on the retention characteristics of the gas chromatography column and the ion mobility of the ion mobility spectrometer, providing a more convenient, faster, and accurate analysis method [19]. Compared to GC-MS detection, HS-GC-IMS technology has a better ability to identify isomers and similar chemical substances [20]. Additionally, samples do not require complex concentration and enrichment, which helps maintain the stability of flavor substances. Therefore, GC-IMS can be widely used to distinguish volatile components and isomers, analyze trace components, and facilitate rapid on-site detection [21].

Currently, research on the chemical composition changes in *Schisandra chinensis* due to processing mainly focuses on liquid-phase components using techniques such as High Performance Liquid Chromatography (HPLC) or Liquid Chromatography–Mass Spectrometry (LC-MS). However, the study of changes and differences in the volatile chemical components and contents of *Schisandra chinensis* processed by different methods remains an unexplored area. This study starts with *Schisandra chinensis* processed by different methods, using steam-processed *Schisandra chinensis* as the control group and vinegar, honey, wine, and salt-processed *Schisandra chinensis* as the experimental groups. It discusses the changes in volatile components and their differential analysis after adding different substances and processing.

## 2. Results and Discussion

### 2.1. Spectrum Analysis

The results detected by the HS-GC-IMS machine were output using VoCal 0.4.03 software and its plugins (Figure 1). The vertical axis of the spectrum represents the gas chromatography retention time of the chemical substances, while the horizontal axis represents the ion migration time of the chemical substances. The red vertical lines in the ion mobility spectrum are the normalized reaction ion peaks (RIP peaks), and the other bright spots in the spectrum are the detected volatile chemical substances. The concentration of the chemical substances can be preliminarily determined based on the color; the redder the color, the higher the concentration of the detected substance. To better observe the differences in volatile substances processed by different methods, steam-processed *Schisandra chinensis* was used as the control group to create differential spectra (Figure 2). The background color of the differential spectrum is nearly white. After subtracting the peaks of the control group, the ion peaks of *Schisandra chinensis* processed by other methods were added. The different colors reflect the concentration differences of the chemical substances. Red peaks indicate that the concentration of the detected volatile chemical substances in the experimental group is higher than that in the control group; blue peaks indicate that the concentration of the detected volatile chemical substances in the experimental group is lower than that in the control group.

### 2.2. Qualitative Analysis of Volatile Chemical Substances

The VoCal software was used to identify and select the ion peaks of the chemical substances detected in the spectra. The selected ion peaks were integrated to calculate the peak volumes, and the different ion peaks were summarized to obtain the volatile component fingerprint profiles of *Schisandra chinensis* processed by different methods (Figure 3). The fingerprint profiles can display the composition of volatile chemical substances in different processed products and the differences in the content of various components. Qualitative analysis was performed using the NIST database included in the software. The chemical substances were identified based on Retention Time and other detected data, resulting in the fingerprint profiles and list of chemical substances for the experimental groups (Table 1) [22].

The qualitative results of volatile chemical substances showed that 125 volatile chemical substances were identified in *Schisandra chinensis* processed by different methods, with an additional 4-methyl-2-pentanol added as an internal standard in the experiment. By combining the monomers and polymers of the same substance for classification, it was found that the experimental samples of *Schisandra chinensis* processed by different methods contained 85 successfully identified volatile chemical substances, including 16 esters, 12 alcohols, 15 ketones, 12 aldehydes, 10 terpenes, 6 olefinic compounds, 2 nitrogen compounds, 2 lactones, 2 pyrazines, 2 sulfur compounds, 2 thiophenes, 1 acid, 1 thiazole, and 2 unclassified chemical substances. Due to the need for further enrichment and supplementation of the database, 15 unknown chemical substances were not identified.

### 2.3. Differential Analysis of Volatile Chemical Substances

#### 2.3.1. Classification Analysis of Chemical Substance Peak Volumes

VoCal software can calculate the peak volume based on the ion reaction peak height and peak area of the selected region. By combining and statistically analyzing the peak volumes of monomers and polymers of the same substance, the total peak volumes of different categories of volatile chemical substances were obtained (Figure 4).

Classification analysis revealed significant differences in the content of volatile chemical substance categories in *Schisandra chinensis* samples processed by different methods: *Schisandra chinensis* processed by vinegar steaming had higher peak volumes of various components than other groups, especially alcohols and acids, which is related to the volatile components of added rice vinegar; *Schisandra chinensis* processed by wine steaming had much higher levels of acid chemical substances than other groups, which is related to the treatment with rice wine. Excluding the abnormally high data from these three groups, it was observed that salt-processed *Schisandra chinensis* contained more aldehydes and the least esters; water-processed *Schisandra chinensis* had the highest content of ketones; pyrazines reached the highest expression after vinegar processing. The five different processing methods of *Schisandra chinensis* showed significant differences in the content of chemical substance categories, with good differentiation, indicating further research value.

#### 2.3.2. PCA (Principal Component Analysis)

Through preliminary data statistics, we found that *Schisandra chinensis* processed by different methods has abundant volatile chemical substances. By summing the peak volumes of chemical substance categories, samples from different groups can be preliminarily distinguished. To further identify the key components of processed *Schisandra chinensis* and the differences in components after different processing methods, we combined the peak volumes of monomers and dimers, obtaining 85 chemical substance peak volumes as dependent variables, with three repetitions of different processing methods as independent variables. PCA (principal component analysis) is a multivariate statistical analysis method often used to discuss the correlation between multiple variables and perform data dimensionality reduction. In the analyzed graph, the differences and similarities between samples can be directly observed through the distances between samples [23,24]. 

After PCA (principal component analysis), it is evident in the image that the processed *Schisandra chinensis* clusters separately (Figure 5). The PCA results show that the PAO2 group differs more significantly from the other four processing methods. The similarity between PAO0, PAO1, PAO3, and PAO4 is high, and PCA cannot directly cluster these four groups, requiring further Orthogonal Partial Least Squares Discriminant Analysis (OPLS-DA).

#### 2.3.3. OPLS-DA Analysis

To further distinguish and analyze the differences in volatile chemical components of *Schisandra chinensis* processed by different methods, OPLS-DA can more accurately differentiate between groups [25] and directly identify characteristic components of different processing methods through subsequent Variable Importance in Projection (*VIP*) value calculations [26].

In the OPLS-DA (Figure 6), we can see that the independent variable fitting indices (R^2^X = 0.921; R^2^Y = 0.988; Q^2^ = 0.749) are all greater than 0.5, indicating an acceptable model fit [27]. To further verify whether the model is overfitted, 200 permutation analyses were performed using the model data (Figure 7). The permutation analysis results showed that the intersection of the Q^2^ regression line with the vertical axis was less than 0, indicating that the analysis was valid.

#### 2.3.4. *VIP* Value Analysis and One-Way Analysis of Variance (ANOVA)

OPLS-DA can establish a model to screen the *VIP* values of various volatile chemical substances, identifying differential variables and locating key differential chemical substances. A *VIP* value > 1 indicates that the chemical substance is a key volatile chemical substance (Figure 8) [28].

SPSS software 27 was used to perform one-way ANOVA on the peak volumes of all detected volatile chemical substances, calculating the significance *P* of all single chemical components. A data table with *P* < 0.05 and *VIP* > 1 was established for further analysis and discussion. After calculation, it was found that 36 chemical substances had *VIP* values greater than 1 (Table 2). Combining with *P* values for joint screening, 24 chemical substances met the screening criteria. The five substances with the highest *VIP* values were Alpha-Farnesene, Methyl acetate,1-octene, Ethyl butanoate, and citral. The *P* values of these substances were all <0.05, indicating significant key differential volatile chemical substances in the experiment.

#### 2.3.5. Heatmap Analysis

After screening for volatile chemical substances with *VIP* > 1 and *P* < 0.05, the peak volumes of the chemical substances were analyzed using the Heatmap plugin in Origin software 2021.

To clearly and identify the correlation between *Schisandra chinensis* processed by different methods and different categories of chemical substances (after screening), the peak volumes of different categories of chemical substances were summed, and the data were normalized using the logarithm log10. The data were then subjected to clustering analysis using Pearson correlation analysis, clustering rows and columns separately. Pearson correlation analysis can be used to evaluate the correlation between different types of volatile chemical substances to explore potential key marker compounds (Figure 9) [29].

In the figure, it can be observed that steam-processed *Schisandra chinensis* is positively correlated with olefinic compounds, with the highest correlation; wine-processed *Schisandra chinensis* is most positively correlated with aldehydes and positively correlated with terpenes, but almost not correlated with thiazole; vinegar-processed *Schisandra chinensis* is positively correlated with various substances, with the highest correlations being sulfur compounds, pyrazines, esters, and acid; honey-processed *Schisandra chinensis* is most positively correlated with thiazoles and most negatively correlated with olefinic compounds; salt-processed *Schisandra chinensis* is positively correlated with ketones and other substances, and most negatively correlated with alcohol. *Schisandra chinensis* processed by different methods shows significant differences in volatile component content and clusters separately.

In the data heatmap (Figure 10), it is evident that *Schisandra chinensis* processed by different methods shows good data separation, and the repeated experiments cluster separately, indicating good repeatability. By adding the peak volumes of individual volatile chemical substances to the analysis, we can directly observe the positive and negative correlations between different processing methods and peak volumes. According to the data model calculations, the key chemical substances screened with *VIP* > 1 and *P* < 0.05 were clustered separately. Vinegar processing was most positively correlated with acetic acid ethyl ester and acetic acid butyl ester; honey processing was most positively correlated with ethyl butanoate and butanal; wine processing was most positively correlated with 1-octene, methyl acetate, and 2-hexenal; salt processing was most positively correlated with 2-methyl-3-ketotetrahydrofuran, 4-methylthiazole, 2(3H)-Furanone, 5-methyl-, and butanal; steam processing was most positively correlated with 1-(2-furanyl) ethanone and hexanal. In the clustering analysis of different processing methods and key volatile chemical substances, it can be seen that honey-processed *Schisandra chinensis* and salt-processed *Schisandra chinensis* cluster together, but their correlation with individual chemical substances is significantly different and can be clearly distinguished. Vinegar-processed *Schisandra chinensis* clusters separately, showing the greatest difference in key differential component content compared to other processed *Schisandra chinensis*.

The above heatmap clustering analysis demonstrates that *Schisandra chinensis* processed by different methods shows significant differences in volatile components based on their similarities and can be accurately clustered separately. The experiments show good repeatability, accurate results, good model fit, and no overfitting.

### 2.4. Methodological Discussion

Currently, most studies on the chemical components of *Schisandra chinensis* focus on the determination of non-volatile organic compounds, while the measurement of volatile components in *Schisandra chinensis* processing for medicinal purposes lacks relevant research. The apparent differences in *Schisandra chinensis* before and after processing cannot be distinguished with the naked eye. This experiment utilized HS-GC-IMS technology to characterize and analyze the differences in volatile chemical components of *Schisandra chinensis* processed by different methods. Key differential components with *VIP* values greater than 1, such as Alpha-Farnesene, Methyl acetate,1-octene, Ethyl butanoate, and citral, were identified in *Schisandra chinensis* processed by different methods. The quantities of these key differential components can be used in the future for rapid identification of the processing methods of *Schisandra chinensis* products. The proposed methodology resolves the difficulty in distinguishing between different processed products of *Schisandra chinensis*; however, it does not fully characterize the component differences in these products. The pharmacological activity of non-volatile organic compounds is significant, and this method does not account for non-volatile substances. Further research is needed to explore the differences in non-volatile organic compound content in *Schisandra chinensis* processed by different methods.

## 3. Material and Methods

The entire process of the Materials and Methods can be simplified into a diagram for easier understanding (Figure 11).

### 3.1. Materials and Experimental Equipment

#### 3.1.1. Experimental Materials

*Schisandra chinensis* was collected from Wangqing and identified by Associate Researcher Peilei Xu from the Institute of Special Animal and Plant Sciences of the Chinese Academy of Agricultural Sciences as Northern Schisandra (*Schisandra chinensis)* of the Magnoliaceae family; edible rice vinegar (Haitian, Zhejiang, China), edible yellow wine (Shaoxing, Zhejiang, China), edible refined honey (Wang’s, Jiangxi, China), analytical pure NaCl (Aladdin, New York, NY, USA), chromatographic pure 4-methyl-2-pentanol (Sigma-Aldrich, St. Louis, MO, USA), and chromatographic methanol (Fisher, Rochester, NY, USA) were also obtained.

#### 3.1.2. Experimental Equipment

FlavourSpec^®^ flavor analyzer; analytical balance (Mettler Toledo, Columbus, OH, USA).

### 3.2. Processing Methods

In 2022, *Schisandra chinensis* was collected from Wangqing County, Jilin Province. After air-drying indoors at 21 °C to a constant weight, it was considered dry *Schisandra chinensis* and vacuum-sealed and stored in a refrigerator at 4 °C. The experimental methods were adapted from those of Zhou at Shenyang Pharmaceutical University, with appropriate modifications for our study.

#### 3.2.1. Steamed *Schisandra chinensis*

Take 100 g of dry *Schisandra chinensis*, remove impurities, rinse it with distilled water, place it in a steamer after the water boils, steam it for 2 h, label it as PAO0, and cool it to 21 °C (room temperature).

#### 3.2.2. Wine-Steamed *Schisandra chinensis*

Take 100 g of dry *Schisandra chinensis* and mix with 20 mL of yellow wine; let it sit briefly, place it in a steamer after the water boils using an induction heater, steam it for 2 h, labeled it as PAO1, and dry it at room temperature until no further weight change occurs after being cooled to 21 °C (room temperature).

#### 3.2.3. Vinegar-Steamed *Schisandra chinensis*

Take 100 g of dry *Schisandra chinensis*, mix it with 15 mL of rice vinegar, place it in a steamer after the water boils using an induction heater, steam it for 2 h, label it as PAO2, and dry it at room temperature until no further weight change occurs after being cooled to 21 °C (room temperature).

#### 3.2.4. Honey-Steamed *Schisandra chinensis*

Take 100 g of dry *Schisandra chinensis*, mix it with 15 g of refined honey, place it in a steamer after the water boils using an induction heater, steam it for 2 h, label it as PAO3, and dry it at room temperature until no further weight change occurs after being cooled to 21 °C (room temperature).

#### 3.2.5. Salt-Steamed *Schisandra chinensis*

Take 100 g of dry *Schisandra chinensis*, mix it with 15 mL of water and 2 g NaCl, place it in a steamer after the water boils using an induction heater, steam it for 2 h, label it as PAO4, and dry it at room temperature until no further weight change occurs after being cooled to 21 °C (room temperature).

### 3.3. Detection Method for Volatile Substances

Grind the *Schisandra chinensis* processed by the above five methods to the same level using a mortar; accurately weigh 0.5 g with an analytical balance, place it in a 20 mL headspace vial, and add 20 μg of 4-methyl-2-pentanol methanol solution with a concentration of 10^−5^ g/mL as an internal standard.

HS-GC-IMS analysis conditions: The experimental instrument is the FlavourSpec^®^ flavor analyzer (Haineng G.A.S company, Qingdao, China). GC-IMS unit analysis time: 45 min. The chromatographic column is a WAX-Columm (15 m long, 0.53 mm inner diameter, 1 μm film thickness; column temperature: 60 °C; carrier/drift gas: N_2_; analysis temperature: 60 °C). Automatic headspace sampling (HS) unit: sample volume: 500 μL; incubation time: 10 min; incubation temperature: 60 °C; injection needle temperature: 85 °C; incubation speed: 500 rpm; analysis temperature: 60 °C. 

Carrier gas flow rate setting: 0–2 min at 2 mL/min; 2–20 min increase in carrier gas flow rate to 100 mL/min; maintain the flow rate until the analysis is completed.

Using the same *Schisandra chinensis* and extraction medium, three independent biological repetitions were performed, with each sample loaded separately (Detailed test conditions and RAW spectral data will be provided in the Appendix A section of the article).

### 3.4. Data Analysis Methods

#### 3.4.1. Preliminary Data Analysis Methods

The preliminary analysis of the detected experimental data was conducted using the specialized data processing software VoCal, equipped with the experimental apparatus. The main functions of the software include the establishment of peak maps, integration calculation of peak volumes, and the creation of fingerprint profiles.

#### 3.4.2. Principal Component Analysis (PCA)

PCA is a mainstream feature extraction algorithm and the most common dimensionality reduction method. It can be used to interpret intergroup differences in multidimensional samples. SIMCA 16.0 software was used to perform dimensionality reduction calculations on the peak volumes of volatile components from different samples, with group settings at A = 3. 

#### 3.4.3. OPLS-DA, *VIP* Value Analysis, and 200-Time Permutation Analysis

SIMCA 16.0 software was used to establish an OPLS-DA model based on the peak volumes of the samples, with group settings at autofit and A = 4 + 5 + 0. The resulting model underwent *VIP* value screening, and a 200-time permutation analysis was performed on all data to verify the model’s validity.

#### 3.4.4. One-Way ANOVA 

The peak volumes of volatile chemical components from the samples were input into IBM SPSS 27 software to conduct one-way ANOVA. The results were presented in the form of *P* values.

#### 3.4.5. Correlation Clustering Analysis

The average peak volumes of volatile components from three repeated detections within the same treatment group were calculated. Depending on the model requirements, peak volumes of the same chemical category were combined, or average peak volumes of volatile chemical compounds with *VIP* > 1 and *P* < 0.05 were screened. The data were normalized using log10 logarithms. After normalization, heatmap clustering was performed using Origin 2021 software, with Pearson correlation analysis as the clustering calculation method. The results were visualized using the www.omicshare.com (accessed on 6 December 2024) platform.

## 4. Conclusions

This study is the first to use HS-GC-IMS technology to detect and compare the volatile components of *Schisandra chinensis* processed by different Traditional Chinese Medicine Processing methods. The fingerprint profiles can accurately distinguish *Schisandra chinensis* processed by different methods. The detection results showed that 125 volatile chemical substances were qualitatively identified in the five different experimental groups, along with 15 unidentified chemical substances. After combining the polymers of the chemical substances, a total of 85 different volatile chemical substances were obtained.

This study used various mathematical analysis methods such as PCA-X, OPLS-DA, one-way ANOVA, and Pearson clustering analysis to establish models and compare and analyze the similarities and differences in the volatile components of *Schisandra chinensis* processed by different methods. By establishing screening models based on *VIP* values and *P* values, 24 key significant volatile chemical substances were identified for processed *Schisandra chinensis*. For different preparation procedures of natural products used for medicinal purposes, it is essential to establish a rapid and accurate non-destructive identification method.

Heatmap clustering analysis was performed based on these substances and the total chemical substance category peak volumes, with peak volumes normalized for Pearson correlation analysis. The analysis results showed significant differences in the chemical composition of *Schisandra chinensis* processed by different methods, which can be clearly distinguished from each other. This study identified the five chemical substances with the highest *VIP* values—Alpha-Farnesene, Methyl acetate,1-octene, Ethyl butanoate, and citral—based on the peak volumes of chemical substances. These substances will serve as marker volatile chemical substances for processed *Schisandra chinensis*, providing a basis for identification and differentiation along with the fingerprint profiles.

## Figures and Tables

**Figure 1 molecules-29-05883-f001:**
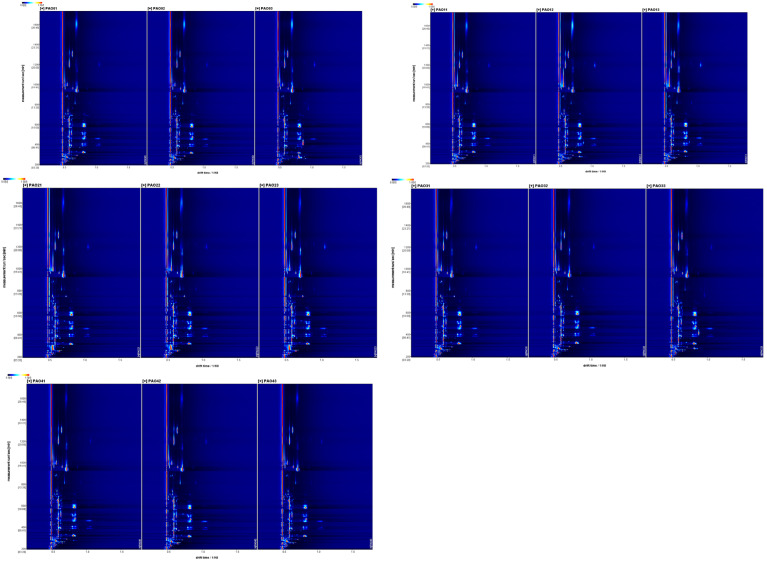
Ion mobility spectra of *Schisandra chinensis* processed by different methods (PAO01–03 for steam processing, PAO11–13 for wine processing, PAO21–23 for vinegar processing, PAO31–33 for honey processing, and PAO41–43 for salt processing; same for the figures below).

**Figure 2 molecules-29-05883-f002:**

Differential spectra of *Schisandra chinensis* processed by different methods compared to steam processing (red indicates increased substances; blue indicates decreased substances).

**Figure 3 molecules-29-05883-f003:**
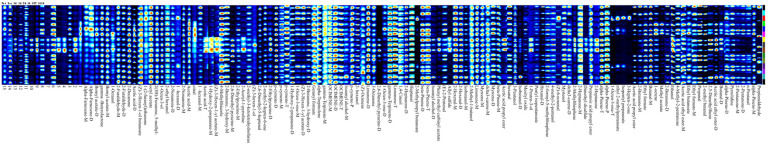
Fingerprint profiles of volatile chemical substances of *Schisandra chinensis* processed by different methods (red and white dots in the fingerprint map represent the peaks of chemical substances detected in the spectrum. Each row of the spectrum represents a different treatment and its replicates, and each column represents a different chemical substance).

**Figure 4 molecules-29-05883-f004:**
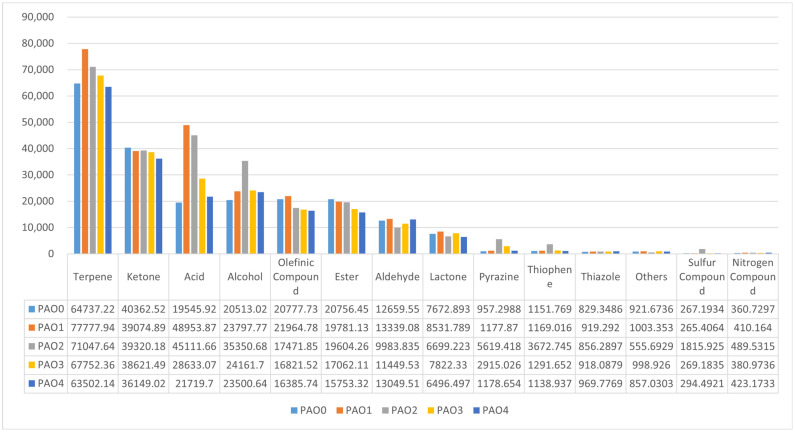
Differences in peak volumes of *Schisandra chinensis* processed by different methods (summed by chemical substance categories). The vertical axis represents the total peak volume, and the horizontal axis represents different categories of chemical substances. The table below provides detailed explanations of the peak volume values represented in the figure.

**Figure 5 molecules-29-05883-f005:**
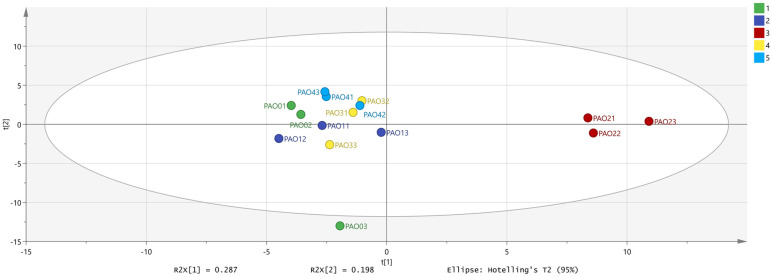
PCA (principal component analysis) of *Schisandra chinensis* processed by different methods (PC1 = 0.287, PC2 = 0.198; the PAO2 group is clearly distinguished from the other groups).

**Figure 6 molecules-29-05883-f006:**
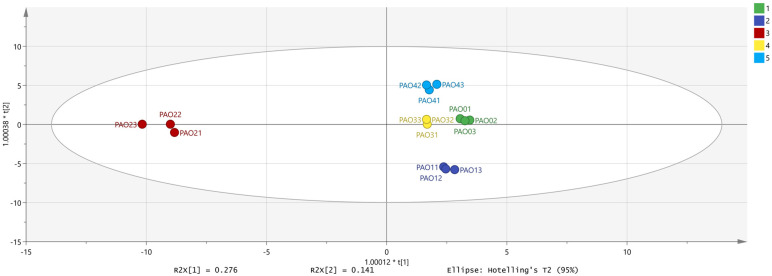
OPLS-DA of *Schisandra chinensis* processed by different methods (R^2^X1 = 0.276, R^2^X2 = 0.141; different groups are clustered separately and are well distinguished).

**Figure 7 molecules-29-05883-f007:**
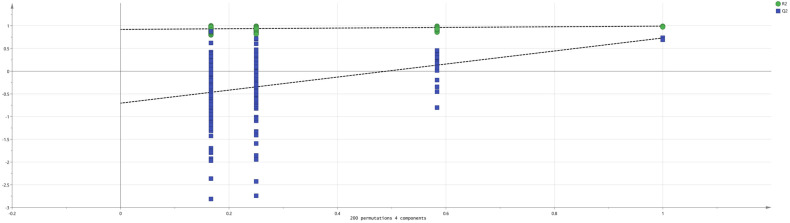
Analysis of 200 permutations.

**Figure 8 molecules-29-05883-f008:**
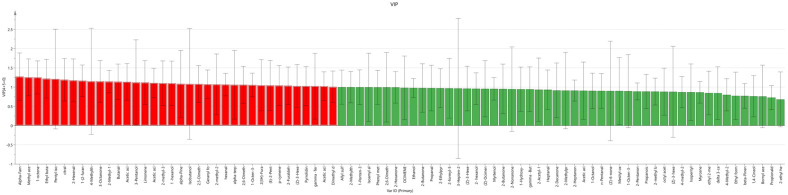
*VIP* value analysis (the red sections represent chemical substances with *VIP* > 1, the green sections represent chemical substances with *VIP* < 1).

**Figure 9 molecules-29-05883-f009:**
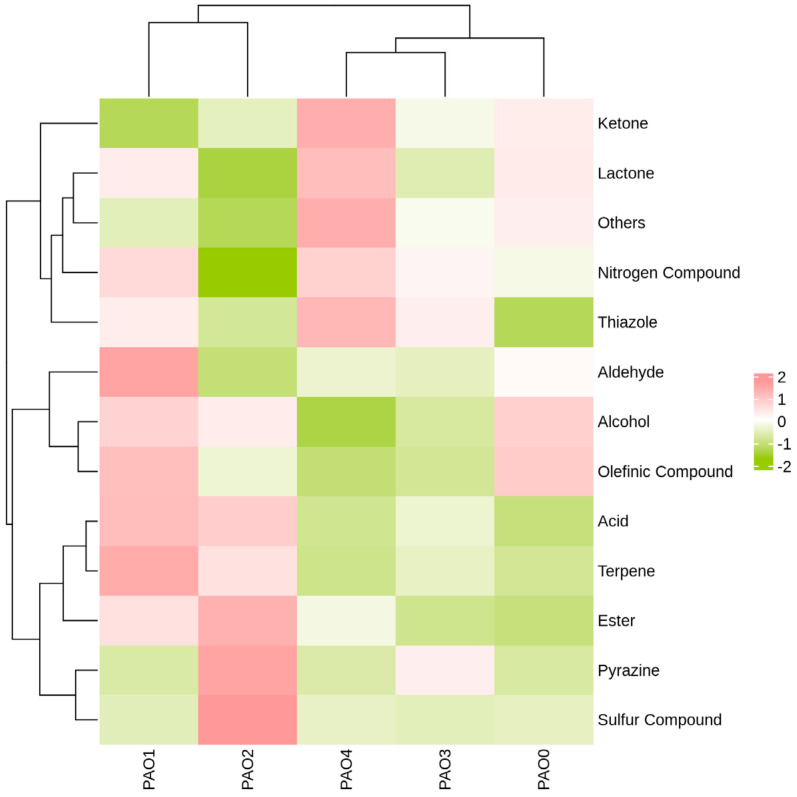
Heatmap clustering of the amounts of different types of volatile chemical substances and different processing methods. (The closer the color of the block is to pink, the higher the Z-score; the closer to olive, the lower the Z-score; and the closer to white, the closer to 0.)

**Figure 10 molecules-29-05883-f010:**
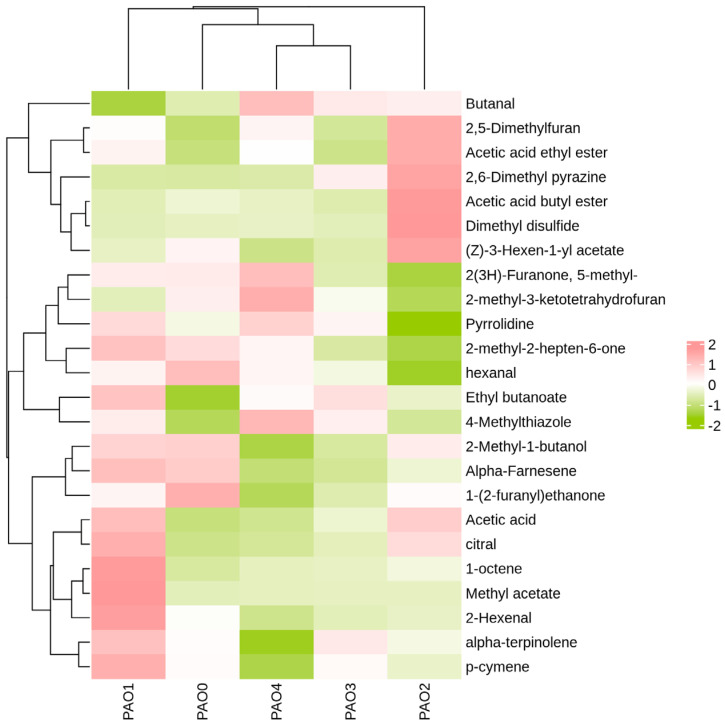
Heatmap clustering of the amounts of different volatile chemical substances and different processing methods. (The closer the color of the block is to pink, the higher the Z-score; the closer to olive, the lower the Z-score; and the closer to white, the closer to 0.)

**Figure 11 molecules-29-05883-f011:**
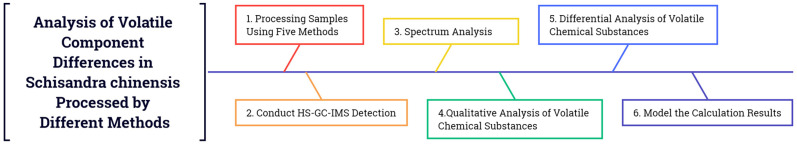
Experimental research diagram.

**Table 1 molecules-29-05883-t001:** List of volatile chemical substances of *Schisandra chinensis* processed by different methods. (In the comments, M stands for monomers of chemical substances, D stands for dimers, T stands for trimers, and P for Polymers. RI for Retention Index, RT for Retention Time, DT for Drift Time.)

Count	Compound	CAS#	Category	MW	RI	RT [s]	DT [a.u.]	Comment
1	Alpha-Farnesene	C502614	Olefinic Compound	204.4	1729.9	1592.47	1.43239	M
2	Alpha-Farnesene	C502614	Olefinic Compound	204.4	1733.1	1602.809	1.45975	D
3	Bornyl acetate	C76493	Terpene	196.3	1591.6	1201.681	1.21588	M
4	Bornyl acetate	C76493	Terpene	196.3	1588.5	1194.162	2.18565	D
5	1-Octanol	C111875	Alcohol	130.2	1578.4	1169.837	1.4701	
6	gamma-Butyrolactone	C96480	Lactone	86.1	1632.4	1305.957	1.31604	
7	2-Furaldehyde	C98011	Aldehyde	96.1	1486	969.453	1.08465	M
8	2-Furaldehyde	C98011	Aldehyde	96.1	1481.8	961.064	1.33283	D
9	Acetic acid	C64197	Acid	60.1	1511.4	1020.842	1.05214	M
10	Acetic acid	C64197	Acid	60.1	1499.5	996.469	1.15284	D
11	2-Decanone	C693549	Ketone	156.3	1473.5	944.998	1.46981	
12	(Z)-3-Hexen-1-ol butanoate	C16491364	Ester	170.3	1471.2	940.538	1.43841	
13	1-(2-furanyl)ethanone	C1192627	Ketone	110.1	1506	1009.703	1.43757	
14	octyl acetate	C112141	Ester	172.3	1469.5	937.334	1.51274	
15	2(3H)-Furanone, 5-methyl-	C591128	Lactone	98.1	1437.6	878.501	1.1238	
16	1-Octen-3-ol	C3391864	Alcohol	128.2	1441.5	885.344	1.16758	
17	2-ethyl hexanol	C104767	Alcohol	130.2	1479.7	957.044	1.79682	
18	2-Nonanone	C821556	Ketone	142.2	1398.3	810.918	1.4062	M
19	citral	C5392405	Terpene	152.2	1660.2	1381.817	1.05235	
20	2-Nonanone	C821556	Ketone	142.2	1399.6	813.097	1.87918	D
21	1-hexanol	C111273	Alcohol	102.2	1369.3	764.385	1.32648	M
22	1-hexanol	C111273	Alcohol	102.2	1361.3	752.049	1.98575	D
23	Acetic acid	C64197	Acid	60.1	1379.4	780.371	1.04326	T
24	1-Hydroxy-2-propanone	C116096	Ketone	74.1	1323.4	696.175	1.04187	
25	(Z)-3-Hexen-1-yl acetate	C3681718	Ester	142.2	1304.6	670.04	1.04198	M
26	4-Methylthiazole	C693958	Thiazole	99.2	1314.6	683.84	1.05121	
27	2-Butanone, 3-hydroxy	C513860	Ketone	88.1	1299.2	662.804	1.05893	M
28	2-Butanone, 3-hydroxy	C513860	Ketone	88.1	1296.7	659.415	1.33055	D
29	2-methyl-3-ketotetrahydrofuran	C3188009	Others	100.1	1281	630.767	1.07369	
30	2,6-Dimethyl pyrazine	C108509	Pyrazine	108.1	1359.9	749.95	1.14014	M
31	2,6-Dimethyl pyrazine	C108509	Pyrazine	108.1	1359.4	749.188	1.53791	D
32	2-Ethylpyrazine	C13925003	Pyrazine	108.1	1321	692.806	1.13081	M
33	2-Ethylpyrazine	C13925003	Pyrazine	108.1	1312.8	681.445	1.15845	D
34	2-Acetyl-1-pyrroline	C85213225	Nitrogen Compound	111.1	1304.7	670.276	1.13019	
35	(Z)-3-hexen-1-ol	C928961	Alcohol	100.2	1391.2	799.339	1.21841	
36	2,6-Dimethyl-5-heptenal	C106729	Aldehyde	140.2	1351.2	736.742	1.17582	
37	2-methyl-2-hepten-6-one	C110930	Ketone	126.2	1328.6	703.666	1.17597	
38	p-cymene	C99876	Terpene	134.2	1289.1	646.579	1.18955	D
39	p-cymene	C99876	Terpene	134.2	1288.1	644.675	1.29912	T
40	p-cymene	C99876	Terpene	134.2	1281.8	632.44	1.16475	M
41	1-Hydroxy-2-propanone	C116096	Ketone	74.1	1313.8	682.714	1.23109	
42	1-Octen-3-one	C4312996	Ketone	126.2	1320.6	692.273	1.26847	
43	alpha-Terpinolene	C586629	Terpene	136.2	1285.1	638.776	1.22435	
44	gamma-Terpinene	C99854	Terpene	136.2	1274.2	617.855	1.21313	M
45	gamma-Terpinene	C99854	Terpene	136.2	1255.1	583.1	1.70573	D
46	p-cymene	C99876	Terpene	134.2	1278.7	626.399	1.18673	P
47	OCIMENE	C13877913	Olefinic Compound	136.2	1255.5	583.868	1.21385	M
48	OCIMENE	C13877913	Olefinic Compound	136.2	1267.4	605.266	1.24943	D
49	Isoamyl alcohol	C123513	Alcohol	88.1	1222.3	527.826	1.24303	M
50	Isoamyl alcohol	C123513	Alcohol	88.1	1221.7	526.879	1.49185	D
51	(Z)-3-Hexen-1-yl acetate	C3681718	Ester	142.2	1320.5	692.192	1.30909	D
52	Geranyl formate	C105862	Ester	182.3	1297.1	659.974	1.21875	
53	OCIMENE	C13877913	Olefinic Compound	136.2	1246.3	567.768	1.25028	T
54	2-hexanol	C626937	Alcohol	102.2	1248.4	571.328	1.28294	
55	(Z)-Ocimene	C470826	Olefinic Compound	154.3	1223.2	529.281	1.2945	
56	1,4-Cineol	C470677	Olefinic Compound	154.3	1215.8	517.403	1.72909	
57	Limonene	C138863	Terpene	136.2	1207.5	504.535	1.29179	D
58	Limonene	C138863	Terpene	136.2	1209.3	507.432	1.65853	T
59	2-Heptanone	C110430	Ketone	114.2	1190.6	479.421	1.72137	
60	3-Octanone	C106683	Ketone	128.2	1261.9	595.194	1.32308	
61	1-Octen-3-one	C4312996	Ketone	126.2	1328.8	703.948	1.69984	
62	Heptanal	C111717	Aldehyde	114.2	1169.7	450.987	1.68416	
63	Phenyl methyl carbinyl acetate	C93925	Ester	164.2	1199.1	491.838	1.04076	
64	2-Methylthiophene	C554143	Thiophene	98.2	1102.1	370.101	1.0448	
65	(E)-2-Pentenal	C1576870	Aldehyde	84.1	1147.3	422.443	1.11402	
66	Allyl sulfide	C592881	Sulfur Compound	114.2	1146.5	421.389	1.13021	
67	Isobutanol	C78831	Alcohol	74.1	1105.8	374.067	1.17171	M
68	Isobutanol	C78831	Alcohol	74.1	1103.4	371.475	1.36373	D
69	2-Hexenal	C6728263	Aldehyde	98.1	1187.9	475.688	1.16633	M
70	2-Hexenal	C6728263	Aldehyde	98.1	1190.1	478.769	1.18993	D
71	2-Methylpropyl butanoate	C539902	Ester	144.2	1141	414.682	1.81094	
72	beta-Pinene	C127913	Terpene	136.2	1121.5	391.699	1.28509	M
73	beta-Pinene	C127913	Terpene	136.2	1133.8	406.006	1.64008	D
74	beta-Pinene	C127913	Terpene	136.2	1131.2	402.979	1.73325	T
75	Myrcene	C123353	Terpene	136.2	1157.9	435.653	1.21677	M
76	Myrcene	C123353	Terpene	136.2	1156.6	434.049	1.29248	D
77	4-methyl-2-pentanol	C108112	Internal standard	102.2	1180	464.726	1.54846	
78	2-Heptanone	C110430	Ketone	114.2	1178.8	463.14	1.64267	
79	1-Penten-3-ol	C616251	Alcohol	86.1	1208.3	505.831	1.34755	
80	Mesityl oxide	C141797	Olefinic Compound	98.1	1145.2	419.816	1.44772	
81	Limonene	C138863	Terpene	136.2	1196.2	487.488	1.21606	M
82	2-Methyl-1-butanol	C137326	Alcohol	88.1	1218.5	521.686	1.21862	
83	delta3-carene	C13466789	Terpene	136.2	1130.1	401.634	1.22054	M
84	3-Pentanol	C584021	Alcohol	88.1	1083.5	353.037	1.21457	
85	Myrtenol	C19894974	Olefinic Compound	152.2	1175.1	458.244	2.16023	
86	delta 3-carene	C13466789	Terpene	136.2	1129.6	401.134	2.18758	D
87	Acetic acid butyl ester	C123864	Ester	116.2	1105.2	373.512	1.24908	
88	hexanal	C66251	Aldehyde	100.2	1099.2	366.986	1.26079	M
89	hexanal	C66251	Aldehyde	100.2	1100.7	368.604	1.56086	D
90	Pentyl isopentanoate	C25415627	Ester	172.3	1101	368.867	1.4703	
91	2-formyl-5-methylthiophene	C13679704	Thiophene	126.2	1114	383.224	1.58825	
92	(Z)-6-nonenal	C2277192	Aldehyde	140.2	1100.8	368.713	1.77005	
93	Dimethyl disulfide	C624920	Sulfur Compound	94.2	1057.2	331.445	1.13176	
94	2-Butanone	C78933	Ketone	72.1	880.8	243.241	1.05576	
95	Ethanol	C64175	Alcohol	46.1	988.2	283.215	1.14115	
96	Acetic acid propyl ester	C109604	Ester	102.1	996.2	286.709	1.47075	
97	n-Pentanal	C110623	Aldehyde	86.1	1005.9	293.016	1.42287	
98	2-Pentanone	C107879	Ketone	86.1	1003.3	291.202	1.36813	M
99	2-Pentanone	C107879	Ketone	86.1	1003.9	291.678	1.39543	D
100	alpha-Pinene	C80568	Terpene	136.2	1026.5	307.94	1.21178	M
101	alpha-Pinene	C80568	Terpene	136.2	1031.1	311.329	1.29451	D
102	alpha-Pinene	C80568	Terpene	136.2	1041.2	318.956	1.67437	T
103	alpha-Pinene	C80568	Terpene	136.2	1037.8	316.408	1.73311	P
104	Propanoic acid propyl ester	C106365	Ester	116.2	1051.2	326.765	1.21666	
105	2-Hexanone	C591786	Ketone	100.2	1054.3	329.179	1.19327	
106	4-Methyl-2-pentanone	C108101	Ketone	100.2	1022.8	305.173	1.17392	
107	1-octen-3-one	C4312996	Ketone	126.2	965	273.178	1.67708	
108	3-Hepten-2-one	C1119444	Ketone	112.2	927.2	257.647	1.63179	
109	ethyl 2-methylpentanoate	C39255328	Ester	144.2	957.7	270.115	1.77095	
110	Isopentyl propanoate	C105680	Ester	144.2	975.1	277.524	1.85323	
111	Pyrrolidine	C123751	Nitrogen Compound	71.1	1007.2	293.963	1.29013	
112	Propanal	C123386	Aldehyde	58.1	826.7	229.208	1.06079	M
113	Ethyl formate	C109944	Aldehyde	74.1	779.9	217.74	1.08903	M
114	Ethyl formate	C109944	Aldehyde	74.1	877.4	242.324	1.07505	D
115	Acetic acid ethyl ester	C141786	Ester	88.1	914.3	252.582	1.10692	M
116	Acetic acid ethyl ester	C141786	Ester	88.1	903.1	249.256	1.33781	D
117	1-octene	C111660	Ester	112.2	845	233.862	1.15705	
118	Propanal	C123386	Olefinic Compound	58.1	807.6	224.455	1.13918	D
119	2-Butanone	C78933	Aldehyde	72.1	922.5	255.798	1.24818	
120	Butanal	C123728	Ketone	72.1	917.5	253.829	1.2913	
121	Methyl acetate	C79209	Aldehyde	74.1	841	232.836	1.19564	
122	Ethyl butanoate	C105544	Ester	116.2	1001.3	289.855	1.21108	
123	2,5-Dimethylfuran	C625865	Ester	96.1	931.7	259.447	1.37262	
124	2-methyl butanal	C96173	Others	86.1	938.5	262.234	1.40162	
125	Propionaldehyde	C123386	Aldehyde	58.1	826.7	229.208	1.06079	M

**Table 2 molecules-29-05883-t002:** *VIP* values and *P* values from ANOVA analysis of *Schisandra chinensis* processed by different methods.

No.	Compounds	*VIP*	*P*	No.	Compounds	*VIP*	*P*
1	Alpha-Farnesene	1.273	0.001	44	OCIMENE	0.983	0.119
2	Methyl acetate	1.254	0.000	45	Ethanol	0.978	0.000
3	1-octene	1.253	0.000	46	2-Butanone, 3-hydroxy	0.977	0.064
4	Ethyl butanoate	1.219	0.022	47	Propanal	0.975	0.247
5	Pentyl isopentanoate	1.210	0.142	48	2-Ethylpyrazine	0.973	0.000
6	citral	1.191	0.000	49	2-formyl-5-methylthiophene	0.969	0.404
7	2-Hexenal	1.175	0.001	50	3-Hepten-2-one	0.966	0.457
8	1-(2-furanyl)ethanone	1.165	0.013	51	(Z)-3-Hexen-1-ol butanoate	0.962	0.122
9	4-Methylthiazole	1.149	0.026	52	2-hexanol	0.959	0.028
10	3-Octanone	1.148	0.165	53	(Z)-Ocimene	0.957	0.163
11	2-Methyl-1-butanol	1.144	0.019	54	Myrtenol	0.956	0.240
12	Butanal	1.140	0.003	55	2-Butanone	0.953	0.322
13	Acetic acid	1.136	0.000	56	2-Nonanone	0.947	0.580
14	3-Pentanol	1.122	0.251	57	1-Hydroxy-2-propanone	0.946	0.020
15	Limonene	1.120	0.142	58	gamma -Butyrolactone	0.945	0.220
16	Acetic acid ethyl ester	1.107	0.004	59	2-Acetyl-1-pyrroline	0.934	0.002
17	2-methyl-3-ketotetrahydrofuran	1.102	0.001	60	Heptanal	0.933	0.343
18	1-hexanol	1.099	0.188	61	2-Decanone	0.915	0.158
19	alpha-Pinene	1.084	0.185	62	2-Methylpropyl butanoate	0.911	0.575
20	Isobutanol	1.080	0.191	63	2-Heptanone	0.910	0.529
21	2,5-Dimethylfuran	1.080	0.013	64	Acetic acid propyl ester	0.908	0.090
22	Geranyl formate	1.072	0.061	65	1-Octanol	0.904	0.136
23	2-methyl-2-hepten-6-one	1.071	0.037	66	n-Pentanal	0.902	0.040
24	hexanal	1.066	0.001	67	(Z)-6-nonenal	0.901	0.596
25	alpha-terpinolene	1.061	0.025	68	Mesityl oxide	0.900	0.310
26	2,6-Dimethyl pyrazine	1.060	0.001	69	1-Octen-3-one	0.897	0.484
27	1-Octen-3-ol	1.056	0.118	70	2-Pentanone	0.888	0.058
28	2(3H)-Furanone, 5-methyl-	1.048	0.034	71	Propanoic acid propyl ester	0.887	0.276
29	(E)-2-Pentenal	1.043	0.137	72	2-methyl butanal	0.885	0.284
30	p-cymene	1.042	0.032	73	octyl acetate	0.882	0.469
31	2-Furaldehyde	1.036	0.079	74	(Z)-3-hexen-1-ol	0.879	0.071
32	(Z)-3-Hexen-1-yl acetate	1.028	0.001	75	Isopentyl propanoate	0.869	0.609
33	Pyrrolidine	1.026	0.004	76	Myrcene	0.865	0.636
34	gamma -Terpinene	1.025	0.110	77	ethyl 2-methylpentanoate	0.846	0.597
35	Acetic acid butyl ester	1.023	0.000	78	delta 3-carene	0.844	0.832
36	Dimethyl disulfide	1.006	0.000	79	4-Methyl-2-pentanone	0.800	0.389
37	Allyl sulfide	1.000	0.000	80	Ethyl formate	0.774	0.972
38	2-Methylthiophene	0.998	0.000	81	beta-Pinene	0.772	0.942
39	1-Penten-3-ol	0.998	0.000	82	1,4-Cineol	0.764	0.857
40	Isoamyl alcohol	0.996	0.128	83	Bornyl acetate	0.759	0.811
41	Phenyl methyl carbinyl acetate	0.996	0.000	84	Propionaldehyde	0.729	0.986
42	2,6-Dimethyl-5-heptenal	0.995	0.064	85	2-ethyl hexanol	0.681	0.471
43	2-Hexanone	0.993	0.000				

## Data Availability

Data are contained within the article and Appendix A.

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
