# Peer review of "Utilizing Headspace–Gas Chromatography–Ion Mobility Spectroscopy Technology to Establish the Volatile Chemical Component Fingerprint Profiles of Schisandra chinensis Processed by Different Preparation Methods and to Perform Differential Analysis of Their Components"

_molecules, 2024, doi:10.3390/molecules29245883_

Round 1

Reviewer 1 Report

Comments and Suggestions for Authors

This study of Dr. Xu and co-authors is the first to use HS-GC-IMS technology to detect and compare the volatile components of Schisandra chinensis processed by different Traditional Chinese Medicine Processing methods. The fingerprint profiles can accurately distinguish Schisandra chinensis processed by different methods. The detection results showed that 135 volatile chemical substances were qualitatively identified in the five different experimental groups, along with 8 unidentified chemical substances. After combining the polymers of the chemical substances, a total of 95 different volatile chemical substances were obtained. The present study used various mathematical analysis methods such as PCA-X, OPLS-DA, one-way ANOVA, and Pearson clustering analysis to establish models, compare, and analyze the similarities and differences in the volatile components of Schisandra chinensis processed by different methods. By establishing screening models based on VIP values and P values, 13 key significant volatile chemical substances were identified for processed Schisandra chinensis. Heatmap clustering analysis was performed based on these substances and the total chemical substance category peak volumes, with peak volumes normalized for Pearson correlation analysis. The analysis results showed significant differences in the chemical composition of Schisandra chinensis processed by different methods, which can be clearly distinguished from each other. The present study identified the five chemical substances with the highest VIP values: Alpha-Farnesene, 1-octene, Methyl acetate, citral, and Ethyl butanoate, based on the peak volumes of chemical substances. These substances will serve as marker volatile chemical substances for processed Schisandra chinensis, providing a basis for identification and differentiation along with the fingerprint profiles. 

The manuscript is relevant for the Section «Natural Products Chemistry» and for Special Issue

«Extraction, Identification and Isolation of Chemical Compounds in Natural Matrices, 2nd Edition» of  “Molecules”. 

The manuscript contains the well-structured Results section, that described volatile chemical components of Schisandra chinensis processed by different Traditional Chinese Medicine Processing methods and establish fingerprint profiles, HS-GC-IMS technology was employed to detect, identify, and analyze Schisandra chinensis processed by five different methods. The manuscript contains good experimental details given in the methods section. 

However, in order to complete the review, it is necessary to make some comments: 

1) Please, change the reference format in the text (Wu et al., 2018; B. Yang et 40 al., 2024) to  reference numbers format. Numbers should be placed in square brackets [ ]. It is very inconvenient to use the references  format that the authors used.

2) Figures 1-3, 8. Please provide high resolution images. The results of the analysis were not readable and analyzed in the presented version.

3) I think that the authors should provide details of the studies using chromatography methods, chromatograms and experimental spectra, etc., as well as details of the calculations in the Supporting information. The article provides only generalizing diagrams and tables. This will significantly improve the perception of the study and simplify the understanding of the material.

The article can be accepted after major revisions.  

Reviewer 2 Report

Comments and Suggestions for Authors

Recommendation: Publish in The Journal of Molecules in its current form.

Comments:
This study provides valuable insights into the volatile chemical fingerprint profiles of Schisandra chinensis using HS-GC-IMS technology, a novel and efficient approach for identifying a wide range of compounds. The authors successfully identified 95 volatile compounds, and their identification is well-supported by spectroscopic data, which appears consistent throughout. The analysis was conducted using several robust statistical methods, including PCA-X, OPLS-DA, and one-way ANOVA, further strengthening the reliability of the findings.

Notably, the study highlights key compounds with the highest VIP scores, such as Alpha-Farnesene, 1-octene, Methyl acetate, Citral, and Ethyl butanoate, based on peak volume analysis. These results contribute significantly to the understanding of Schisandra chinensis's chemical composition and are relevant to the broader natural products scientific community.

Given the novel use of HS-GC-IMS and the thorough analytical approach, I support the acceptance of this manuscript in The Journal of Molecules.

Reviewer 3 Report

Comments and Suggestions for Authors

The manuscript identifies chemical compounds after different extractions of Schisandra chinensis which in itself is a research topic relevant to extract identification. However, in many regards the manuscript fails scientific standards; the “best” part are “Results” and “Conclusions” since they stick to their points, the rest of the manuscript is deficient or missing.

The introduction in its chemical part is more or less suitable, the health claim part has to be rejected.

The methods section appears partly to be copied out of a lab book. Normally, the methods section describes what was done, not what should have been done; also, modifications or special occurrences are mentioned in this section. It also mentions peculiarities like „which method was used to remove impurities“, whereas in line 308 the manuscript mentions „partially based on! – which part was changed, and in which way; how long is „briefly“? Also, extraction media are not specified (yellow wine, rice vinegar, refined honey” are no extraction media, but have to be specified. The description of the detection method is unusable. As a rule, a methods section repeated in another lab should result in exactly the same method and result which cannot be obtained in the present form! A most important information from the methods section is the number of repetition; since this is not given, and the Figures are mere printouts I have to believe that the assays were done as singular determinations. In the results section „three rpetitions of different processing methods“ are mentioned, but not explained whether these repetitions are done for sampling plants, the complete workup or only three independent analyses of one extract.

In the results no mention is made for identification on individual compounds, or the logic the internal NIST database uses. At least the second part of the Results appear to be a reproduction of the functions inherent in the FlavorSpec flavor analyzer, not a scientific study let alone a discussion of the relevance and comparability of the results..

The legends to both Tables and Figures are incomplete, e.g. they lack explanations of abbreviations used. E.g. in Table 1: what is indicated by M, D or T? In Figure 4, AUC differences are presented. Since no number of repetition is given the Figure seems to represent single repetitions; in this case no error bars can be given, also no differences between extraction methods are proven. Figure 5 is a principal component analysis; I can identify a maximum of 15 dots for 5 groups; doing elliptiuc approximations may mathematically be possible but does not make much scientific sense. Could it also be circles with a much wider overlap, or lines with uncertainty deviations? And since the discussion is included in the text, why does the cinegar extract differ from the other ones?

A discussion is missing; in the section “results and discussion” no mention is made of comparable studies or results. I am no expert in metabolomics but even my rudimentary knowledge knows that just running the possibilities of a machine over a data set isn't enough to create relevant or valid data. Results have to be interpreted which is not done in the manuscript since any discussion is lacking (both for pitfalls in the methods used as well as in the results they present).

Mentioning unexplained abbreviations in a manuscript may be possible and is at the discretion of the authors; however, rarely used or specialist abbreviations or acronyms must be explained. Thus: what is OPLY-DA analyses, what is VIP value analysis? Explaining this at least reduces confusion.

The Literature section is a mess, the style arbitrary. Nearly all citations are from Chinese authors although both methods and analysis are used worldwide. Chinese language citations are not accessible or readable by reviewers and ought to be avoided. Even in the text, bad examples of citations are present, e.g. in line 46 – what is meant ny n.d.?

All together, the manuscript has no scientific hypothesis or standard. For a publication the manuscript has to be completely rewritten; the introduction must stick to the research data without fantasizing about unwarranted claims, the methods have to be completely described so they may be reproduced independently, its data have to be discussed both in terms or correctness (possible sources of error), its relevance (can it be used for the purposes specified in the hypothesis) and context (how does it fit with other results). None is done, therefore a reject.

A few selected specific items:

lines 44 - 66: Health claims are either not supported, founded on cell culture experiments, or the sources not accessible by non-Chinese readers. Therefore for me, no claim has been proven! Consequently, the claim that the extracts are health-promoting, also is not supported!

Lines 117ff: A reviewer should at least in theory be familiar with a method and its result output. This paragraph reads like an introduction for students, not researchers using this or similar chromatographic methods.

Line 124: „To more intuitively observed“ - Intuition is not scientific; a first impression has to be supported by quantitative readouts.

Lines 134 – 140, Figures 1 and 2: even when magnifying to 400%, I can't identify many differences but the pictures go blurry. In this presentation the pictures do not prove anything, except – on manification analysis – the lack of differences between the samples.

Lines 150 – 152: see comments to Figure 1 and 2, here also the authors do not describe the results (maybe even indicate some selected compounds like their five important ones by arrows) and present the results as a blurry or wavy figure. No emntion is made where the compounds listed in Table 1 are represented in Fig. 1 – 3, nor by which method the identity was proven.

Line 237: “Combining with P values for joint screening” - how were p values for joint screening corrected? It has to be done but the methods section has to identify the method used! Also, which screening criteria were used?

Comments on the Quality of English Language

I doubt that the authors really understand the meaning of some English words like "intuitively". Although the English grammar and style are fine, the wording is sometimes cumbersome or wrong.

Reviewer 4 Report

Comments and Suggestions for Authors

The manuscript titled "Utilizing HS-GC-IMS Technology to Establish the Volatile Chemical Component Fingerprint Profiles of Schisandra Chinensis Processed by Different Preparation Methods and Perform Differential Analysis of Their Components" presents a novel and exciting approach for identifying volatile phytochemicals in mixtures obtained through various preparation methods. The study conducted by the authors investigates the volatile chemical compounds of Schisandra chinensis processed by methods other than traditional Chinese medicine, using HS-GC-IMS  technology. In total, 95 volatile compounds were identified, including esters, alcohols, ketones, and terpenes, among others. Analyses such as Pearson correlation, OPLS-DA, and PCA were applied to evaluate and classify the detected substances.

The results showed significant differences in the chemical profiles depending on the processing method used. Five prominent compounds were identified: alpha-farnesene, 1-octene, methyl acetate, citral, and ethyl butanoate, which are considered valuable markers for the future identification of processed Schisandra chinensis. This type of analysis not only helps to characterize the plant but is also essential to ensure quality and authenticity in its use in traditional medicine. However, the following issues need to be overcome.

Major revisions:

·         A diagram showing the steps of this research would guide the reader through each part of the research. Please include it.

Many abbreviations and acronyms need to be defined.

·         In the methods section, it is important to include the software and parameters of how each test was performed (PCA, OPLS-DA, and Pearson correlation analysis methods). Although it is briefly mentioned in the results, nothing is mentioned in the methodology. Please include this information.

·        Mention a section in the methodology of how each fingerprint profile was obtained.

·         All figures, in addition to the caption, must contain a brief explanation of what is depicted, as well as information about the most outstanding aspects. Please include it.

·   A Venn diagram of the ordinary and non-common compounds between the compounds obtained in steam-processed Schisandra chinensis as the control group and vinegar, honey, wine, and salt-processed Schisandra chinensis as the experimental groups might provide more context for the reader.

·         The citation style is incorrect; please pass each reference using the MDPI style.

·         It is important to list the components of each fingerprint profile analyzed in this work because it is difficult to identify them in the figures. Please include these results and the others (i.e., PCA) in supplementary material.     

·    Figures 1, 2, and 3 cannot be read, so you need to improve the resolution to 300 dpi.

·         In the methodology used, the authors should perform an in-silico prediction to obtain the fingerprint profile of the compounds obtained and also include them within their model (PCA, OPLS-DA, and Pearson correlation analysis methods) to be able to make a comparison of the results obtained by their methods and those predicted by the in-silico analysis. This analysis could allow a comparison of what was obtained experimentally with what was predicted in silico, allowing us to generate information and say that if we cannot obtain a fingerprint profile experimentally, just having the compounds could. It helps researchers to have a surrogate in silico model to compare the compounds obtained by the different extraction methods.

·        The authors should consider discussing the limitations of their methodology, which solely involves analyzing the fingerprints of volatile products and does not encompass preparations that contain non-volatile phytochemicals. Their work is important, as this approach is valuable for assessing effective preparations because of the presence of volatile compounds. However, the impact of the preparation may be underestimated when non-volatile active ingredients are also included. Your work highlights these limitations and paves the way for further research.

·         In the conclusion section, the authors should highlight the contributions of the the methodology used to compare preparation procedures for natural products intended for medicinal use.

·         It is recommended that authors further explain the potential applications of the newly discovered substances (Alpha-farnesene, 1-octene, methyl acetate, citral, and ethyl butanoate) as markers.

Comments on the Quality of English Language

Minor revisions:

Comments on the Quality of English Language: There are some typographical mistakes.

Round 2

Reviewer 1 Report

Comments and Suggestions for Authors

Thanks authors for revisions. Article can be published after minor revision, because References list  is still not very useful because it is not fully formatted, many articles do not have pages or article numbers, for example: 

27. Yang, X. Characterization of the Effect of Different Cooking Methods on Volatile Compounds in Fish Cakes Using a Combination of GC–MS and GC-IMS. Food Chemistry 2024. 

28. Li, Z. Comparative Key Aroma Compounds and Sensory Correlations of Aromatic Coconut Water Varieties: Insights from GC ×GC-O-TOF-MS, E-Nose, and Sensory Analysis. Food Chemistry 2024

Reviewer 3 Report

Comments and Suggestions for Authors

I am sorry but the manuscript still has to be rejected. The authors still raise medical claims without support (see below); their Methods section still reads like a manual, the Figures mostly cannot be identified, and error bars are given but no SD values. The software models were originally not developed for peak analysis in GC-MS but partly for metabolomics, partly for expression analysis. Using these models for other purposes as is done in the manuscript needs a good justification, and a discussion of pitfalls in this usage outside of the developmental area.

The text is littered with unspecific terms even when quantitative data esist as indicated by error bars in Figures; “More” is mentioned although it is not clear whether this “more” is significant or not. By the way using error bars for triplicates is possible but only makes mathematical sense, not chemical sense.

Many technical questions remain due to the fact that the method section does not describe what was done but how it should be done. Critical references (in a form readable for non-Chinese readers) must be supplied, whereas many references from the literature may be unnecessary – I can’t say which since the literature section is a mess and often omits crucial information like volume and pages, sometimes even Journal and year are missing. Thus the manuscript does not meet scientific standards, and the shortcomings in its presentation make the content nearly unjudgeable.

To be clear I would welcome a paper which identifies methods to clearly separates extraction methods, supplied with quantitative data on relevant compounds. Unproven claims about their value in any use must be omitted, a careful and sensible speculation about what the data might be used for is acceptable.

I list my complaints more specifically; most of these points are absolute musts for a scientific paper.

lines 148 – 152: Since the citations are incomplete I cannot verify the claims made in these lines, these are invalid, no medical use has been shown by the authors or cited!

The next two sentences needs a citation but have none.

Citation 12: Only the abstract is available online, the “free full text” bounces (DOI Not Found This DOI cannot be found in the DOI System)

Citation 13: I cannot find this article, in the Internet it is not present.

Lines 379 – 388: There still is no scientific hypothesis; listing an “unexplored area” and discussing “changes in volatile components and their differential analysis” is no hypothesis but a statement. What was the purpose of the research? Include more components, widening a spectrum? It cannot be a correlation between volatile components and in vivo effects since these are not investigated.

Lines 392 – 399: this reads like an extract from the handbook; alternatively, it is an introduction to GC analysis for lay people.

Line 399: “intuitively” is no scientific term; it is ok to use intuition as a starting point for formulating a hypothesis, and then investigating this, but no tool for data interpretation.

Lines 401 – 408: see above, reads like a handbook copy.

Figures 1, 2 and 3: The figures are so small, and after magnification blurry that I cannot interpret the data. If I cannot confirm the authors conclusions I cannot judge the quality of the research.

Lines 463 – 473: ok; but it is trivial that unknown peaks are not identified!

Lines 474 – 479: ok

Figure 4: The authors do not note how often a specific sample was analyzed, or even better, how often a specific extraction procedure was performed, and how many replication samples were processed (3 repetitions are mentioned in line 537). They indicate error bars, so it must be more than one analysis, but I miss a statistical analysis of these SD values. In the text (lines 517 - 528), only phrases like “much more” or “much” are used, but this falls short of exact work. A scientific analysis must include whether differences are significant or not (and I am not yet talking about relevant).

Lines 517 – 542: the description has no merit. Scientifically, a quantitative approach must be taken, especially with quantitative data. Instead, the authors only use comparators: more, less, “significant differences” (without statistics!).

Figure 5: a plot from the software is shown; however, no description exists for the parameters creating the uncertainty ares (ellipses). At least for PA01 and PA03 other ellipses are possible still containing all data points.

Figure 6: Obviously, the same data are analyzed as in Figure 5; now the numbers run from 1 to 5, instead of 00 to 04. Please, be consistent.

Figure 8: Again a picture which cannot be read. No mention is made about the rror bars, which in some examples will be negative in the minimum.

Lines 712 – 769: This paragraph is a mixture of hypothesis, aim of he study and questions; a manuscript should address these points and answer the questions quantitatively (if possible). Thus: What are the differences between the extraction process (finding acetic acid in vinegar extracts hardly is surprising); how do the key differential components identify the extracts from each other, as well as from extracts with other plants; how do they correlate with pharmacological effects (I doubt it, and the authors have not shown any proof)? Some of the shortcomings of this article which I addressed in my first review and which have not been addressed.

Figure 11 – not necessary. The methods section must describe the material and methods in a way that a knowledgeable reviewer or a colleague intent on repeating the work can judge whether the research has been conducted properly, and whether there are possible pitfalls.

Lines 780 – 784: this is no regular text.

Lines 791: “were partially based on the experimental methods of Zhou from Shenyang Pharmaceutical University.” -- in this case the differences have to be described, and the original work must be cited.

Lines 793ff, lines 841ff, lines 846, lines 851 etc: This no description of what has been done, but rather a copy from a standard operation procedure. The methods section requires what has been done, not what should be done!! A standard method would read eg.: “10 g S. chinensis were homogenized using a mortar and pistil until a homogenous powder was obtained. The powder was added to 25 ml vinegar and stirred for 6h at room temperature. Afterwards the slurry was filtered using a paper filter grade 3HW; the clear liquid was stored at 4°C until analysis.” This is just a suggestion but should indicate how a methods section should describe how an extract (in this case) has been obtained. If the narrative style of the methods section is similar in other related papers than none of this is in accordance with scientific standards.

line 875: Three independent repetitions – which steps were done in parallel, and which were done from the same sample? Were three samples of S. chinensis fruits separately extracted etc., or were three aliquots analyzed in the HS-GC-IMS?

Lines 878 - 912: For the software packages it is important what they can and cannot achive. Most of these programs are written for metabolomics, not for extracts, and therefore are not necessarily correct for the use in VOC analysis. Whether or not some chemometric parameters are affected is irrelevant, if a tool is used outside its original usage.

Conclusions: The only acceptable paragraph in the manuscript. No mention of TCM use, strict reduction to the data the manuscript really supplies: 5 (or 13) compounds that can indicate the extraction method of S. chinensis.

References; This part is a total mess. Only 9 out of 30 citations have a doi (and some are invalid – I tested no. 13), likely only 2 articles from non-Chinese authors, many citations only with the publication year, and sometimes not even mention of the Journal!!

Comments on the Quality of English Language

I still consider "intuition" a word indicating: "I have a feeling that the data do show an important point, but I do not not why this should be." Science must be based on reproducible methods and quantitative data, intuition may be used for generating afirst hypothesis.

Reviewer 4 Report

Comments and Suggestions for Authors

The manuscript titled "Utilizing HS-GC-IMS Technology to Establish the Volatile Chemical Component Fingerprint Profiles of Schisandra chinensis Processed by Different Preparation Methods and Perform Differential Analysis of Their Components" presents the acquisition of the fingerprint profile of compounds obtained from five extraction methods of volatile phytochemicals in a plant represents a groundbreaking and captivating approach. The authors present results that achieve the goals of the research.

In this revised manuscript, the authors have provided more comprehensive details regarding the introduction, methodology, and discussion. The following suggestions may further improve the clarity and richness of the manuscript.

According to response number 1. Presenting a Fishbone diagram is an adequate skill. Adding more text boxes could enhance detail and provide a complete overview of the manuscript's contents.

Regarding answer 6, relating the sets of substances identified in the five separation procedures using a Venn diagram could be more complex. However, the methodology employed to determine the fingerprints allows for a quick and reliable qualitative analysis. The compounds identified by the databases in this investigation could be grouped and distinguished based on their frequency of occurrence in each extraction method. It would benefit the reader to summarize the common and uncommon substances in each preparation to associate them with potential therapeutic properties. This preliminary knowledge can be obtained without quantifying and validating the substances. I agree that quantification and standardization are important. However, this could set the tone as a preliminary result for future research.

Regarding answer 8, raw data could be added to the supplementary data, such as data related to statistical analysis, such as PCA, OPLS-DA, and Pearson correlation analysis methods. These data would facilitate the interpretation of the manuscript figures and reproducibility in future research.

Regarding answer 12, I understand and am sorry about this reviewer. I appreciate your sincerity.

So, addressing all minor revisions, I think this manuscript is suitable for publication.

Comments on the Quality of English Language

 Minor editing of the English language are required.
